# Amikacin or Vancomycin Exposure Alters the Postnatal Serum Creatinine Dynamics in Extreme Low Birth Weight Neonates

**DOI:** 10.3390/ijerph18020662

**Published:** 2021-01-14

**Authors:** Tamara van Donge, Anne Smits, John van den Anker, Karel Allegaert

**Affiliations:** 1Pediatric Pharmacology and Pharmacometrics, University Children’s Hospital Basel (UKBB), University of Basel, 4001 Basel, Switzerland; tamara.vandonge@ukbb.ch (T.v.D.); JVandena@childrensnational.org (J.v.d.A.); 2Department of Development and Regeneration, KU Leuven, 3000 Leuven, Belgium; anne.smits@uzleuven.be; 3Neonatal Intensive Care Unit, University Hospitals Leuven, 3000 Leuven, Belgium; 4Division of Clinical Pharmacology, Children’s National Health Hospital, Washington, DC 20010, USA; 5Department of Pharmaceutical and Pharmacological Sciences, KU Leuven, 3000 Leuven, Belgium; 6Department of Hospital Pharmacy, Erasmus MC University Medical Center, 3015 Rotterdam, The Netherlands

**Keywords:** creatinine, vancomycin, amikacin, renal impairment, acute kidney injury, adverse drug reaction

## Abstract

Background: Disentangling renal adverse drug reactions from confounders remains a major challenge to assess causality and severity in neonates, with additional limitations related to the available tools (modified Kidney Disease Improving Global Outcome, or Division of Microbiology and Infectious Diseases pediatric toxicity table). Vancomycin and amikacin are nephrotoxic while still often prescribed in neonates. We selected these compounds to assess their impact on creatinine dynamics as a sensitive tool to detect a renal impairment signal. Methods: A recently developed dynamical model that characterized serum creatinine concentrations of 217 extremely low birth weight (<1000 g, ELBW) neonates (4036 observations) was enhanced with data on vancomycin and/or amikacin exposure to identify a potential effect of antibiotic exposure by nonlinear mixed-effects modelling. Results: Seventy-seven percent of ELBW patients were exposed to either vancomycin or amikacin. Antibiotic exposure resulted in a modest increase in serum creatinine and a transient decrease in creatinine clearance. The serum creatinine increase was dependent on gestational age, illustrated by a decrease with 56% in difference in serum creatinine between a 24 or 32-week old neonate, when exposed in the 3rd week after birth. Conclusions: A previously described model was used to explore and quantify the impact of amikacin or vancomycin exposure on creatinine dynamics. Such tools serve to explore minor changes, or compare minor differences between treatment modalities.

## 1. Introduction

### 1.1. How to Recognize the Signal in the Noise

There are numerous reasons why approval of drugs for use in (pre)term neonates is still lagging behind. A major issue is the more difficult demonstration of efficacy and safety, as this commonly needs assessment tools tailored to this population [1,2,3]. Related to safety and adverse events, this includes assessment of seriousness, causality, and severity within the framework of regulatory requirements [1,3]. For both causality as well as severity, disentangling adverse drug reactions from confounders (associated morbidity, multi-pharmacotherapy, and maturational changes in laboratory values) remains challenging [1,3], and this results in inconsistency in the documentation of safety-related issues to, e.g., data monitoring committees or authorities [1,4]. In an effort to improve standardization of adverse event severity, a neonatal adverse event severity scale has recently been developed within the International Neonatal Consortium (INC) [5]. Adverse event severity in neonates was defined by the effect of an event on age-appropriate behavior, basal physiologic functions, and care changes in response to the adverse event, and was applied to 35 common neonatal adverse events (neurological, respiratory, cardiovascular, gastrointestinal, infectious, or general). In this initial version, adverse events related to laboratory values were explicitly not yet included because reliable reference values for (pre)term neonates were not available [5].

Very recently, we reported on serum creatinine dynamics in the first 6 weeks of postnatal life in a relevant cohort with 2814 creatinine observations in 148 extreme low birth weight (<1000 g, ELBW) neonates [6]. Besides maturational changes (gestational age, postnatal age), caesarean section, and ibuprofen exposure, neither exposure to inotropic agents nor prenatal lung maturation (betamethasone) affected these dynamics, in line with previous observations [7,8].

The same dataset and approach can be used to assess the incidence and severity of renal impairment for other commonly used medicines in neonates [9]. As recently described by Al-Turkait et al. in this special issue on drug utilization studies in pediatrics, both aminoglycosides and vancomycin are in the top 10 of most commonly prescribed drugs in neonates, and are nephrotoxic [10]. We, therefore, selected these compounds to assess their impact on creatinine dynamics as a likely more sensitive tool to detect and quantify a signal of renal impairment.

### 1.2. Aminoglycosides and Vancomycin Related Acute Kidney Injury: Developmental Toxicology

Toxicity has limited the use of aminoglycosides, but there is a consistently lower rate of acute kidney injury (AKI) in neonates when compared to adults [11]. This suggests either developmental toxicology (less in neonatal life), or poorer performance to recognize the signal in neonates (inaccurate AKI tool), or both. Renal tubular cell toxicity starts with binding to acidic phospholipids of the cell membrane (aminoglycosides are charged negative), followed by cellular uptake, facilitated by megalin and cubulin ligands [11]. These ligands-related uptakes display saturation, explaining the practice of extended time interval dosing of aminoglycosides to avoid toxicity, and improve efficacy (peak concentration related) [9,11,12]. Intracellular accumulation in endosomes and mitochondrial stress subsequently occurs (related to ribosomal inhibition, similar to the bacterial ribosomal inhibition). Because of the maturational expression of these ligands, it seems that aminoglycosides are much better tolerated in neonates compared to adults. In neonates, Nestaas et al. reported an incidence of 8.4% for nephrotoxicity (increased creatinine, urinary aminopeptidase, 50/589 events) in neonates exposed to aminoglycosides [13].

Vancomycin exposure is also associated with a higher risk of AKI in adult critical care, with a relative risk of 2.45 (95% confidence interval, 1.69 to 3.55) [14]. Based on both in vitro and in vivo observations, the assumed mechanism also relates to dose-dependent intracellular accumulation and oxidative damage in the renal tubular cells [15]. In neonates, the evidence on vancomycin nephrotoxicity has been summarized and is suggested to occur in 1–9% of exposed cases [16,17,18]. These incidences are supported by two more recent reports [19,20]. In the Leroux prospective validation study, only 2/190 cases displayed nephrotoxicity (either 2-fold increase, or +0.6 mg/dL in creatinine over 48 h after start), both associated with hemodynamic instability and vasopressor need, and with subsequent normalization [19], while Germovsek et al. observed nephrotoxicity (neonatal Kidney Disease, Improving Global Outcomes (KDIGO) definition) in 4/77 neonates [20]. Target attainment for vancomycin exposure (AUC_24h_ > 400) was common in both recent cohorts, indicating that these findings better reflect contemporary target exposure. This is of relevance, as toxicity relates to the exposure (AUC_24h_ > 800) and the duration of vancomycin administration (cumulative exposure, AUC_cum_) [15,21]. Finally, there are to the very best of our knowledge only a limited number of observations on renal impairment during simultaneous administration of aminoglycosides and vancomycin in neonates [21,22,23].

A key issue is hereby the limitations of the currently available tools to qualify and quantify renal impairment or AKI in (pre)term neonates [24,25]. Almost a decade ago, Jetton and Askenazi suggested an adapted AKI definition (neonatal modified KDIGO) specific for use in neonates [24]. In essence, the definition is based on trends (increase) in serum creatinine concentrations and in urine output to result in AKI staging (stage 0–3) (Table 1).

However, as the dynamics in serum creatinine are most pronounced in ELBW neonates with an initial physiological increase up to day 3 and a subsequent progressive decrease over the next weeks, these trends are difficult to capture within a ‘fixed or proportional increase to former creatinine observation’ approach, and perhaps does not provide the granularity needed to detect ‘minor changes’ [25]. This absence of sufficient granularity to capture the complex dynamics of serum creatinine in (pre)term neonates is even more prominent when considering the Division of Microbiology and Infectious Diseases (DMID) pediatric toxicity table specific to creatinine. Although relevant to the assessment of antimicrobial agents (Table 2) in drug registration studies, there is only one single cut off value to grade creatinine adverse events, only driven by postnatal age (in the first week of life, day 7–60, and day 61–90, respectively) [26].

The aim of this study was to quantify the effect of aminoglycoside and/or vancomycin exposure on renal impairment reflected by the change in serum creatinine dynamics during the neonatal period of ELBW neonates. This knowledge may inform and help clinicians to recognize and examine the potential of renal impairment in their vulnerable patients beyond the commonly used AKI definition.

## 2. Materials and Methods

### 2.1. Study Population, Clinical Characteristics and Ethics

The dataset consists of a pooled dataset of 4026 serum creatinine observations in 217 ELBW cases in the first 42 days after birth and admitted at the Neonatal Intensive Care Unit of UZ Leuven, as recently published [6]. Serum creatinine was analyzed enzymatically by Roche (Roche Diagnostics, Mannheim, Germany) in all cases, and all measurements were isotope dilution mass spectrometry (IDMS) traceable. All clinical data (demographics, but also information on co-medication like ibuprofen or inotropic agents and type of delivery) were already available in this dataset [6].

For the purpose of this project, all individual electronic medical files were searched for days exposed to vancomycin, amikacin, or both, as this is the established antibiotic regimen for suspected late onset sepsis (LOS) in the Leuven neonatal intensive care unit (NICU) [27]. Vancomycin and amikacin were administered according to hospital guidelines, as previously published [11,28].

Subsequent adaptations of this empiric treatment were made by the attending neonatologist, and were based on the pathogen isolated and the clinical assessment [29]. Another indication for amikacin in this NICU, of relevance for the current study, is necrotizing enterocolitis (co-treatment with piperacillin-tazobactam). Ethical approval of the current study covered the additional data search (model development and evaluation, S63405) [6].

### 2.2. Refined Serum Creatinine Model

The recently developed and published model that characterized the serum creatinine concentrations for ELBW infants was applied and further strengthened with data on individual administration of vancomycin or amikacin exposure to identify a potential effect of antibiotic (AB) exposure on individual serum creatinine concentration profiles and their related creatinine clearance values [6]. The respective mean population parameters and inter-individual variability were attained by nonlinear mixed-effects modelling analysis (Appendix A). The original model was developed in 148 ELBW neonates from which 2814 serum creatinine concentrations were collected during the first six weeks after birth and validated with a population of 69 ELBW neonates with 1212 serum creatinine concentrations. In this previous study, the elimination rate of creatinine was characterized as the ratio between creatinine clearance (Equation (1)) and volume of distribution.
(1)$$CL\left(t\right)=\;CL_{BL}+\frac{emax\;\times \;t^{Hill}}{t_{50}{}^{Hill}+\;t^{Hill}},$$
where *t* reflects postnatal age (days), *CL_BL_* reflects baseline creatinine clearance (L/day), emax is the maximum additional achieved clearance (L/day), t_50_ corresponds to the time point (in days) where half of emax is achieved, and the Hill coefficient describes the steepness of the creatinine-time relationship. To include the physiological and crucial weight changes during the early weeks of life, volume of distribution was based on the current weight measurements. Gestational age (GA) and mode of delivery showed an association with postnatal maturation of the creatinine clearance (faster clearance increase with advancing GA and after C-section). Additional details on the dynamic model can be found in the original paper [6].

In the current study, the potential effects of AB exposure on creatinine clearance (and corresponding serum creatinine concentrations) for each postnatal day were identified as (i) only vancomycin exposure, (ii) only amikacin exposure, (iii) either vancomycin or amikacin exposure, and (iv) combined vancomycin and amikacin exposure. These four conditions were investigated by applying a stepwise forward selection and backward deletion approach based on the likelihood ratio test (*p* < 0.05) to identify significant covariate-parameter relationships. Model evaluation was performed by precision of the estimated parameters (residual standard error, RSE), the maximization of the likelihood (decrease of objective function value (OFV) of at least 3.84 points for one additional model parameter), goodness-of-fit plots (observed versus predicted creatinine concentrations), and visual predictive checks. Software package Monolix (version 2019R1. Antony, France: Lixoft SAS, 2020, http://lixoft.com/products/monolix/) was used to fit individual data to the mathematical model. Data handling, graphical visualization and numerical calculations were performed in R (version 3.5.1; R Development Core Team, Vienna, Austria, http://rproject.org).

To illustrate the effect of AB exposure on serum creatinine profiles in ELBW cases with different gestational ages, model-based simulations, incorporating covariate relationships, were performed in typical ELBW neonates, stratified for four gestational ages, namely of 24, 27, 29, and 32 weeks. The antibiotic exposure duration comprised of three constituent days (clinical practice for suspected LOS) and was set to start on day 5 (first week of life), day 19 (third week of life), and day 33 (fifth week of life) after birth. Comparison was performed between the predicted serum creatinine and creatinine clearance over a period of six weeks for typical ELBW cases who were either or not exposed to AB. Simulations were performed in a deterministic setting where no inter-individual variability is considered and the predicted median profiles were illustrated as a function of included covariates. Additionally, stochastic simulations, including inter-individual variability, were carried out to assess the difference in predicted creatinine concentrations between patients being exposed to AB exposure and those who are not. In total, 1000 simulations were performed for each gestational age group.

## 3. Results

### 3.1. Population

The median gestational age and birth weight was 27 weeks (26–28 weeks IQR) and 830 g (720–910 g interquartile range (IQR)), respectively (Table 3). Of the studied ELBW population, 72% and 71% received (at least one day) vancomycin or amikacin during the first six weeks of life, respectively. Of these patients, 77% were exposed to either vancomycin or amikacin, and 66% were simultaneously exposed to vancomycin and amikacin for at least one day (Table 3). During the first six weeks after birth, patients were exposed on average 6.8 days (0–10 days IQR) and 5.6 days (0–9 days IQR) to vancomycin or amikacin, respectively. Patients with a lower gestational age were more often exposed to AB, as compared to patients with a higher gestational age (Figure 1).

### 3.2. Refined Serum Creatinine Model

Antibiotic exposure, defined as receiving either vancomycin or amikacin, resulted in a lower overall creatinine clearance and higher serum creatinine concentrations due to a significant increase of 30% in t_50_ when exposed to AB (−40.6 points in OFV). The parameter t_50_ reflects the time point in days where half of the maximum additional achieved clearance is reached. A longer t_50_ corresponds with lower creatinine clearance values. The combination of vancomycin and amikacin exposure did not indicate a synergistic effect on the clearance parameters. For a typical ELBW patient of 24 weeks with or without being exposed to AB, the t_50_ was estimated at 31.9 days and 41.7 days, respectively, and decreased with approximately 10% for each increased week of gestation (Table 4).

Predicted serum creatinine concentration time profiles for four typical ELBW neonates are shown in Figure 2a. An increase in serum creatinine concentrations is observed during AB exposure. Figure 2b mirrors this, as it illustrates increasing creatinine clearance during the postnatal period, depending on gestational age and represents the diminished clearance capacities when exposed to AB.

The extent of the difference in predicted serum creatinine concentrations during AB exposure at the end of first week of life is less pronounced when focusing on various gestational age groups (Figure 3). This distinction is more obvious for AB exposure at the end of the third and fifth week of life, with a higher difference in serum creatinine concentrations for the youngest ELBW neonates. For exposure at the end of the first week of life, the difference between being exposed to AB or not was on average 0.056 mg/dL (Table 5). For exposure at the end of the third week, i.e., starting on day 19 until day 21 after birth, the difference in creatinine concentration for a neonate at 24 weeks gestation amounts 0.05 mg/dL, and decreased with 56% for a 32-week-old neonate. For a typical neonate of 24 weeks gestation, the creatinine clearance at day 21 after birth was estimated to be 0.44 mL/min being exposed to AB and increased with 12% to 0.49 mL/min without AB exposure. A similar trend was observed for exposure at the end of the fifth week, i.e., from day 33 to 35 after birth, where the difference in creatinine concentrations between being exposed and not exposed, decreased with 60% between 24 and 32 weeks of gestation (Table 5).

## 4. Discussion

This study illustrates that antibiotic exposure, defined as either vancomycin or amikacin treatment, is associated with an increase in serum creatinine concentrations due to a decreased creatinine clearance capacity in the ELBW neonatal population. These changes can be quantified, but the increase in creatinine generally remained below the 0.3 mg/dL threshold of the modified KDIGO tool, as the most recently developed AKI tool (Table 1 vs. Table 5). These increases are even less reflected when applying the (DMID) pediatric toxicity table specific to creatinine (Table 2 vs. Table 5) [26]. This suggests that more tailored, sensitive tools need to be used to fully quantify the incidence and extent of renal side effects of drugs, like amikacin, vancomycin, or the earlier reported ibuprofen, and these may serve for a tailored or comparative pharmacovigilance [6]. Of notice is that the absolute increase in serum creatinine in ELBW cases depends on additional characteristics, further illustrating the difficulty to ‘retrieve the signal’ based on (fixed) increments only [25].

The increase in serum creatinine due to antibiotic treatment is most pronounced when administered during the first week of life (day 5 to 7 after birth), as a result of limited clearance capacities due to immature kidney function [30]. This increase of serum creatinine concentrations during the first week of life appears to be independent of gestational age. When focusing on the difference in serum creatinine concentrations during the third week after birth and fifth week after birth, a gestational age dependency is observed (Figure 3). This is because a higher increase in serum creatinine is observed for the ELBW patient with a lower gestational age. An increase of 0.05 mg/dL or 0.02 mg/dL is observed for an ELBW neonate of 24 weeks gestational or 32 weeks gestation, respectively, when antibiotic exposure occurs from day 19 to day 21 after birth.

With the current dataset, it was not possible to characterize the potential synergistic effect of combination therapy of vancomycin and amikacin. This is most likely due to the fact that the days that ELBW neonates are exposed to either vancomycin or amikacin (77%) is almost similar to the days that these neonates are exposed to both antibiotics simultaneously (66%), as the combination is the standard treatment for LOS (Figure 1) [27]. The estimated effect of receiving either vancomycin or amikacin on t_50_, one of the parameters characterizing creatinine clearance, increased with 30% when exposed to an antibiotic, resulting in a lower creatinine clearance. This effect did not significantly differ (27%) when assessing the combination therapy or improved the model fit. Interestingly, Salerno et al. very recently reported on the association between nephrotoxic drugs (including aminoglycosides, i.e., gentamicin or tobramycin, and vancomycin) and AKI, using the modified KDIGO criteria in a larger cohort of preterm (22–36 weeks) cases [31]. Seventeen percent of cases in this dataset were classified with at least stage 1 AKI (Table 1). However, based on Odd ratios with the combined exposure to gentamicin + indomethacin as reference, duration of therapy (days) and sepsis, but not combined exposure to aminoglycosides + vancomycin, it was associated with an increased odd of AKI [31]. Related to this analysis, we would like to repeat that the need for inotropic agents (as surrogate marker for sepsis) was not an independent covariate in our model.

The small but quantifiable fluctuations observed in creatinine concentrations and creatinine clearance can be identified as a limitation, but this study showed that the impact of both maturational and non-maturational related effects-like exposure to antibiotics on kidney function-can be recognized, including renal adverse drug reactions. Recognition of such drug toxicity signals in neonates, especially considering the vulnerable ELBW population, should be further developed, as we illustrated its feasibility.

## 5. Conclusions

A previously described model on creatinine dynamics has been used to explore and quantify the impact of exposure to amikacin or vancomycin on creatinine dynamics. It seems that such a tool is more sensitive to detect the signal of renal impairment beyond the existing AKI grading system. In this way, such tools can be used to explore minor changes, or compare minor differences between treatment modalities.

## Figures and Tables

**Figure 1 ijerph-18-00662-f001:**
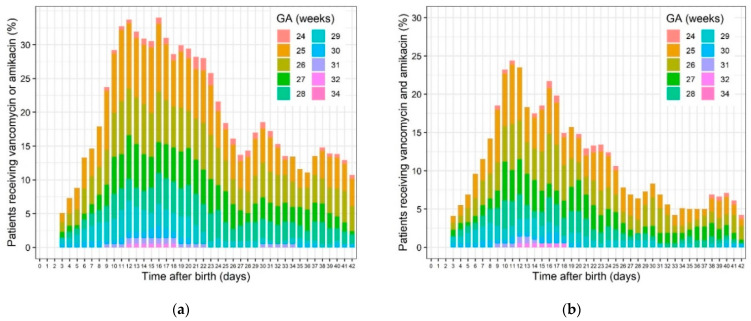
Percentage of extreme low birth weight (ELBW, <1000 g) cases (different gestational age, GA) receiving (**a**) vancomycin or amikacin exposure and (**b**) simultaneous vancomycin and amikacin exposure for each postnatal day.

**Figure 2 ijerph-18-00662-f002:**
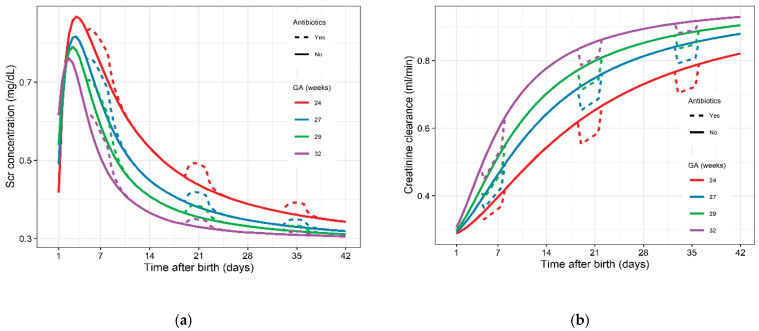
(**a**) Predicted serum creatinine concentration time profiles for four typical GA ELBW neonates; dotted lines represent serum creatinine concentration under AB exposure, solid lines represent absence of AB exposure. (**b**) Predicted creatinine clearance under AB exposure; dotted lines represent creatinine clearance under AB exposure, solid lines represent the absence of AB exposure. GA, gestational age; ELBW, extremely low birth weight; AB, antibiotic.

**Figure 3 ijerph-18-00662-f003:**
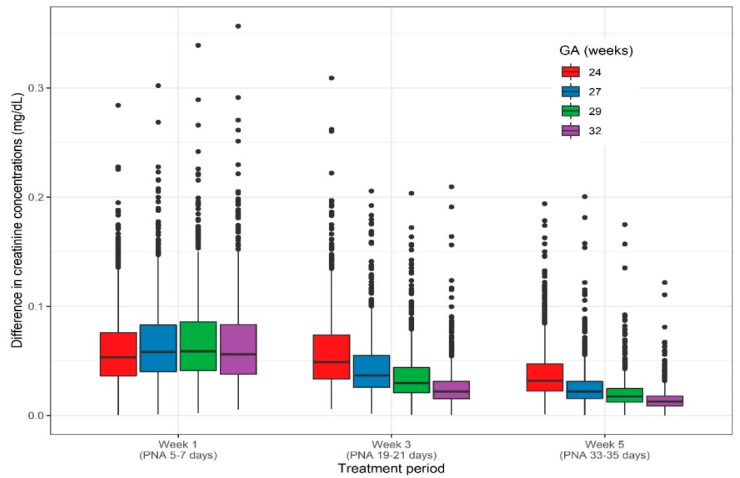
Predicted differences in creatinine concentrations (mg/dL) calculated at the end of each AB exposure period, starting at different postnatal ages and grouped per gestational age. Differences are based on 1000 individual simulation for each gestational age group, including inter-individual variability.

**Table 1 ijerph-18-00662-t001:** Definition of neonatal Acute Kidney Injury (AKI) by serum creatinine (S_cr_) and urine output [24].

Stage	Serum Creatinine (S_cr_)	Urine Output
0	No change in S_cr_ or ↑ < 0.3 mg/dL	>1 mL/kg/h
1	S_cr_ ↑ ≥ 0.3 mg/dL within 48 h or S_cr_ ↑ ≥ 1.5–1.9-fold change vs. S_cr_ * within 7 days	>0.5 and ≤1 mL/kg/h
2	S_cr_ ↑≥ 2 to 2.9 vs. reference S_cr_ *	>0.3 and ≤0.5 mL/kg/h
3	S_cr_ ↑≥ 3-fold change reference S_cr_ * or S_cr_ ≥ 2.5 mg/dL ** or dialysis	≤0.3 mL/kg/h

* Reference S_cr_ is the lowest prior S_cr_ measurement; **, a S_cr_ ≥ of 2.5 mg/dL in neonates reflects an estimated glomerular filtration rate <10 mL/min/1.73 m^2^_,_ ↑ = increase.

**Table 2 ijerph-18-00662-t002:** Creatinine ranges as suggested to grade adverse event severity (grade 1–4) in the Division of Microbiology and Infectious Diseases (DMID) pediatric toxicity table [26].

Postnatal Age	Grade 1	Grade 2	Grade 3	Grade 4
<7 days	1.0–1.7 mg/dL	1.8–2.4 mg/dL	2.5–3.0 mg/dL	>3 mg/dL
7–60 days	0.5–0.9 mg/dL	1.0–1.4 mg/dL	1.5–2.0 mg/dL	>2 mg/dL
61–90 days	0.6–0.8 mg/dL	0.9–1.1 mg/dL	1.2–1.5 mg/dL	>1.5 mg/dl

**Table 3 ijerph-18-00662-t003:** Demographic data presented as median [interquartile range] or as number (%); obs, observations.

Patient Characteristics	
ELBW neonates (obs.)	217 (4026)
Serum creatinine obs. per patient	19 [14, 24]
Gestational age (weeks)	27 [26, 28]
Birth weight (g)	830 [720, 910]
Current weight (g)	920 [790, 1130]
obs. per patient	5 [4, 6]
Antibiotic exposure (yes)	
Vancomycin	72%
Amikacin	71%
Vancomycin or amikacin	77%
Vancomycin and amikacin	66%
Ibuprofen exposure (yes)	133 (61%)
Mode of delivery	
Vaginal	70 (32%)
Cesarean	147 (68%)
Sex	
Female	115 (53%)
Male	102 (47%)
Neonatal death	28 (13%)

**Table 4 ijerph-18-00662-t004:** Parameter estimates of refined creatinine model together with effect size estimates. CV, coefficient of variation; MOD, mode of delivery (1 for C-section and 0 for vaginal delivery); AB, antibiotic exposure present (1 for yes and 0 for no); IBU, ibuprofen exposure (1 for yes and 0 for no). Crea_birth_, creatinine concentration at birth; Kin_production_, production rate of creatinine; *CL_BL_*, baseline creatinine clearance; Emax, maximum additional achieved clearance; t50, timepoint where half of Emax is achieved; Hill, hill coefficient. Median gestational age (GA) was set at 26.56 weeks.

Parameter (Unit)	Estimates (RSE%)	IIV (CV%)
**Population parameters**		
Crea_birth_ (mg/dL)	0.631 (1.95%)	0.256 (26%)
Kin_production_ (mg/day)	4.11 (1.07%)	-
*CL_BL_* (L/day)	0.0798 (2.98%)	0.274 (28%)
Emax (day-1)	0.995 (3.87%)	-
t_50_ (days)	22.8 (7.76%)	0.449 (47%)
Hill	1.31 (4.76%)	0.476 (50%)
**Covariate parameters**		
GA effect on Crea_birth_	1.35 (21.1%)	$Crea_{birth_{i}}=Crea_{birth_{pop}\times {\frac{GA_{i}}{GA_{m}}}^{1.35}}$
IBU effect on *CL_BL_* (no/yes)	3.3 (1.36%)/3.23 (1.85%)	$CL_{BL_{i}}=CL_{BL_{pop}}\times \left(1+3.3\right)\;or\;CL_{BL_{pop}}\times \left(1+3.23\right)$
GA effect on t_50_	−3.09 (19%)	$t_{50_{i}}=t_{50_{pop}}\times {\frac{GA_{i}}{GA_{m}}}^{-3.09}\times \left(1-0.271\times MOD\right)\times \left(1+0.305\times AB\right)$
MOD effect on t_50_	−0.271 (30.3%)	
AB effect on t_50_ (yes)	0.305 (0.91%)	
**Residual variability**		
Proportional error	0.109 (1.27%)	(11%)

**Table 5 ijerph-18-00662-t005:** Differences in serum creatinine concentration (mg/dL) calculated at the end of each period of exposure for three antibiotic periods separated per gestational age group and starting at different postnatal ages. Data presented as median and interquartile range (25th and 75th percentiles).

Gestational Age	Differences in Creatinine Concentration (mg/dL)		
	Week 1 (Day 5–7)	Week 3 (Day 19–21)	Week 5 (Day 33–35)
24 weeks	0.05 (0.04–0.08)	0.05 (0.03–0.07)	0.03 (0.02–0.05)
27 weeks	0.06 (0.04–0.08)	0.04 (0.03–0.5)	0.02 (0.02–0.03)
29 weeks	0.06 (0.04–0.09)	0.03 (0.02–0.04)	0.02 (0.01–0.02)
32 weeks	0.06 (0.04–0.08)	0.02 (0.01–0.03)	0.01 (0.01–0.01)

## Data Availability

The data available on request from the authors (karel.allegaert@uzleuven.be). The data are not publicly available due to restrictions related to privacy.

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
