# Peer review of "Amikacin or Vancomycin Exposure Alters the Postnatal Serum Creatinine Dynamics in Extreme Low Birth Weight Neonates"

_ijerph, 2021, doi:10.3390/ijerph18020662_

Round 1

Reviewer 1 Report

van Donge et al. develop a mathematical model capable of explaining more precisely the impact of exposure to amikacin or vancomycin on creatinine dynamics. The work is well planned, designed and the results are presented appropriately. In addition, it includes supplementary material that explains in detail the statistical concept underlying the methodology used. However, I consider that it is necessary to improve the following aspects:

- Major corrections:
The Discussion section is basically a redescription of the results presented in different words. They do not compare their results with those obtained in similar works by other authors, so they do not include any reference (except one of themselves). This section should be greatly improved before its final publication.

It is necessary to explain in the legend of Table 3 all the abbreviations used, since it is impossible to understand what is presented in this table.

- Minor corrections:
Page 3 (line 10 of section 2.1): define the abbreviation NICU (this is the first time it appears in the text).
Page 4 (line 5 of paragraph 2): why do two question marks appear after 'settings' and 'conditions'?
Page 4 (line 1 of section 3.1): define the IQR abbreviation (this is the first time it appears in the text).
Table 4: All numbers must be presented with the same number of significant figures.

Author Response

van Donge et al. develop a mathematical model capable of explaining more precisely the impact of exposure to amikacin or vancomycin on creatinine dynamics. The work is well planned, designed and the results are presented appropriately. In addition, it includes supplementary material that explains in detail the statistical concept underlying the methodology used. However, I consider that it is necessary to improve the following aspects:

  1. Major corrections: The Discussion section is basically a redescription of the results presented in different words. They do not compare their results with those obtained in similar works by other authors, so they do not include any reference (except one of themselves). This section should be greatly improved before its final publication.

We accept your comment and reflected on this, but still wanted to focus on the fact that the tools used (and their validity, and clinical ‘relevance’) should be further explored in (pre)term neonates. We therefore deliberately further added details on the tools applied in similar studies in the introduction part of the paper, and have added also this in the abstract introduction.

Furthermore, we have added some additional reflections on the limitations of the currently available tools in the discussion section of the paper, and a very recent (2021, Salerno et al) paper that focused on odd ratio’s for acute kidney injury (AKI) for combined exposure to aminoglycosides and vancomycin

  1. It is necessary to explain in the legend of Table 3 all the abbreviations used, since it is impossible to understand what is presented in this table.

We thank the reviewer for highlighting this. The abbreviations used in Table 3 are explained in the legend of the table and in addition, these are also available in the text (Method section 2.2).

  1. Minor corrections: Page 3 (line 10 of section 2.1): define the abbreviation NICU (this is the first time it appears in the text).
    Added to the text.
  2. Page 4 (line 5 of paragraph 2): why do two question marks appear after 'settings' and 'conditions'?
    Sentence is adapted to “these four conditions”.
  3. Page 4 (line 1 of section 3.1): define the IQR abbreviation (this is the first time it appears in the text).
    Abbreviation defined and added to text.
  4. Table 4: All numbers must be presented with the same number of significant figures.
    Numbers in table 4 are adjusted accordingly.

Reviewer 2 Report

In the present study, the authors have analyzed the effect of aminoglycoside and/or vancomycin exposure on renal impairment during the neonatal period of ELBW neonates. In this regard, this study provides some interesting and important information. However, I have several comments as follows.

  1. There is no explanation of the abbreviation ELBW in the Title/Abstract, but it appears in the text, so it should be explained where it first appears.
  2. The authors should revise the text where the question mark appears; “These four settings? conditions? were investigated by applying a stepwise forward selection and back-ward deletion approach based on the likelihood ratio test (p < 0.05) to identify significant covariate-parameter relationships”.(p4 line14)
  3. The authors need to show the amount and duration of antibiotic use. The details of their reasons for the use of antibiotics should be also addressed.
  4. Are there any other nephrotoxic drugs used in combination or during the neonatal period?

Author Response

In the present study, the authors have analyzed the effect of aminoglycoside and/or vancomycin exposure on renal impairment during the neonatal period of ELBW neonates. In this regard, this study provides some interesting and important information. However, I have several comments as follows.

  1. There is no explanation of the abbreviation ELBW in the Title/Abstract, but it appears in the text, so it should be explained where it first appears.

The abbreviation of ELBW is added to the abstract (still respecting the word count) and in the introduction (second paragraph), and the full version has been added to the title.

  1. The authors should revise the text where the question mark appears; “These four settings? conditions? were investigated by applying a stepwise forward selection and back-ward deletion approach based on the likelihood ratio test (p < 0.05) to identify significant covariate-parameter relationships”.(p4 line14)

Sentence is adapted to “these four conditions”.

  1. The authors need to show the amount and duration of antibiotic use. The details of their reasons for the use of antibiotics should be also addressed.

In the first hours to days of treatment for a late onset sepsis, the primary focus is to deliver an effective treatment in a setting of suspected infection. During this earliest stage, mortality is directly related to the effects of the life-threating infection and managing its toxicity is less central. Of note, the causative organism generally remains unknown and treatment duration is most often not exceeding 3-4 days. We have added some additional information on this in the relevant methods section (2.1.)

On the specific quantification requests: Figure 1 illustrates the percentage of neonates receiving vancomycin and/or amikacin per day after birth. On average days, neonates were exposed to 6 days (0-10 days IQR) of vancomycin and 5 days (0-9 days IQR) of amikacin, during a period of 6 weeks. This information is added to the result section (Section 3.1). As requested, information on the amount (dose and interval for amikacin and vancomycin respectively) have been added.

  1. Are there any other nephrotoxic drugs used in combination or during the neonatal period?

In this neonatal population we have investigated exposure to ibuprofen, inotropic agents and antibiotics such as vancomycin and amikacin. In the initial model assessment, it was shown that ibuprofen treatment decreases the clearance of creatinine (albeit to a small extent). Treatment with inotropic agents (as surrogate marker for sepsis) or maternal betamethasone treatment did not impact clearance of creatinine. This has been added to the introduction.

In the current study we illustrate that the extension of the earlier developed model, with additional information on antibiotic (combination) treatment, showed that antibiotic treatment impacted the clearance of creatinine, and therefore affected the renal function.

Round 2

Reviewer 1 Report

All the suggestions have been taken into account and all the errors solved, so I accept the publication of the article.

Reviewer 2 Report

The manuscript has been revised well.